

# Optimal power allocation for a wireless cooperative network with UAV

Jing Yan[1,2], Xuefeng Deng[2], Jihua Liu[3] and Liming Wang[1]

[1] National Key Laboratory of Electronic Test Technology, North University of China, Taiyuan, Shanxi, China
[2] School of Information Science and Engineering, Shanxi Agricultural University, Taigu, Shanxi, China
[3] Department of Computer Science, Lvliang University, Lvliang, Shanxi, China

## ABSTRACT

The main objective of this work was to investigate the optimal power allocation strategy in the UAV cooperative wireless Decode and Forward (DF) relay network. Firstly, the outage probability of the system with and without diversity gain was derived. Two optimization problems were studied for different application scenarios. One of the optimization problems sought to determine an optimal power allocation strategy in certain total power constraint to minimize the system outage probability. Since the optimization problem we established is convex, the Lagrange multiplier method was adopted. For the system without diversity, the explicit expression of the optimal power allocation was derived. The relationship between UAV transmission power and source node transmit power was obtained for the diversity gain system and then the Newton iterative method was used to obtain the optimal power allocation method. The simulation results show that the optimal power allocation strategy can reduce the outage probability of the system under the same conditions, and the reliability of the system was improved. Another optimization problem aimed to use the lowest power to ensure that the outage probability within a certain specific threshold for saving energy resources. Because the optimization problem is non-convex, we proposed an effective method to solve the optimal power allocation strategy. Similarly, we derived the closed-form solution of the power allocation strategy for the system without diversity. Finally, the simulation results verify the correctness of the proposed algorithm.

Corresponding authors
Jing Yan, yanjing_sxau@163.com
Liming Wang, wlm@nuc.edu.cn

## INTRODUCTION

The direct link between communication nodes will become unreliable with increasing space distance among nodes due to the influence of the multipath effect and shadow fading of the environment in the wireless network communication. To improve the performance (coverage, throughput, outage probability, and bit error rate) of wireless communication networks (cellular network, WLAN, and WSN), relay cooperative transmission technology has attracted extensive attention (*Alemayehu & Kim, 2017*; *Egashira et al., 2017*; *Indrasen & Pratap, 2018*; *Katla & Babu, 2020*; *Liu, Zhang & Leung, 2012*; *Tutuncuoglu, Varan & Yener, 2015*).

With the development of technology and the gradual reduction of manufacturing cost, UAV has gradually developed from the initial military equipment to civil consumer products, resulting in a large number of new applications, including a meteorological monitoring system, forest fire-prevention technology, man-machine field, *etc.* (*Chen, 2020*; *Jiao et al., 2019*; *Ma et al., 2018*). It is generally considered that the introduction of unmanned units to build a new wireless network architecture is the key technology to establish an integrated ubiquitous network in the future. UAVs can be deployed in various environments on a large scale according to the requirement, to reduce the cost of network deployment. In addition, its flexible mobility and reasonable flight path planning can effectively improve the performance of wireless networks. Wireless power supply communication network is considered as a promising future communication network architecture, which can satisfy the energy and communication needs of the nodes in the Internet of things at the same time. Wireless energy transmission technology is one of the research hotspots in wireless energy supply communication networks. The great idea of wireless energy transmission technology has been put forward by a famous electrical engineer Nikola Tesla for a long time. A large number of scientists have studied this subject for over a century, but there has been no major improvement. Notably, the research team of MIT finally made a great breakthrough in wireless energy transmission technology until 2007. They applied the principle of electromagnetic coupling resonance to wireless energy transmission and lit a 60-watt bulb with a power supply two meters away through wireless energy transmission (*Aristeidis, Joannopoulos & Marin, 2008*; *Kurs et al., 2007*). The breakthrough of wireless energy transmission technology provides us with a new direction of energy balance strategy, which can realize energy balance through wireless transmission from the nodes with high-energy to low-energy nodes. Establishing UAV assisted wireless energy transmission system which uses UAV as an airmobile energy transmitter to supply power to ground nodes is considered to be an effective solution to improve energy transmission efficiency (*Baek, Han & Han, 2020*; *Yuan et al., 2021*). The team from Michigan State University has built a UAV wireless energy transmission system to provide wireless energy for sensor nodes successfully (*Mittleider, Griffin & Detweiler, 2016*). The research shows that the near-field wireless energy transmission based on magnetic resonance coupling (*Griffin & Detweiler, 2012*; *Mittleider, Griffin & Detweiler, 2016*) and the far-field wireless energy transmission based on RF signal (*Baek, Han & Han, 2020*; *Xu, Zeng & Zhang, 2018*) can be used to provide wireless energy for low-power nodes such as sensors. These works further verify the feasibility of UAVs in the application of wireless energy transmission systems. Furthermore, the UAV flight trajectory optimization algorithms in different scenarios are proposed to improve energy transmission efficiency for UAV wireless energy transmission systems (*Feng et al., 2020*; *Yang et al., 2019*). The network of a UAV equipped with a directional antenna is studied where two ground nodes are charged through wireless power transmission technology, and the balance between the received energy intensity of the two ground nodes is revealed by optimizing the deployment position of the UAV (*Wu, Qiu & Xu, 2018*).

The network where the UAV acts as a relay node has also received wide attention, and the relevant research work mainly on the following aspects: the optimal layout of

UAV relay, flight path planning, and network performance analysis (*Baek & Lim, 2019*; *Lhazmir et al., 2019*; *Nguyen et al., 2020*). The research on the performance of UAV relay communication systems mainly focuses on the performances, such as bit error rate, receiver signal-to-noise ratio, outage probability, capacity, and so on. Most researchers analyze the performance of relay transmission systems from the perspective of the physical layer (*Abualhaol & Matalgah, 2011*; *Jiang & Swindlehurst, 2010*; *Zhan, Swindlehurst & Lee, 2011*). The features of the link between relay UAV and ground receiver under fading channel is analyzed, and then the analytical expression of performance is deduced. Additionally, the optimization algorithm is proposed for relay UAVs based on a certain performance optimization criterion. During the era of the increasing shortage of natural resources, reducing communication energy consumption has become a key point of the communication industry. In the wireless relay system, the research on power consumption is essential, as the optimal allocation can significantly improve the energy efficiency of the system if the power of each node is limited. A wide range of research work has been conducted on power allocation algorithms to improve the energy efficiency of the wireless communication network. The joint resource allocation problem is investigated in a multicast scenario with simultaneously information decode and energy harvest, where the UAV sends information and wireless energy to multiple users on the ground, and then the user decodes and collects energy based on the power split (PS) receiver structure (*Kang & Chun, 2020*). *Zhang et al. (2017)* considered the network where the UAV is regarded as an AF relay node and optimized the trajectory, transmission power, and mobile device power of the UAV node to minimize the outage probability of the system.

*Song et al. (2019)* studied a full-duplex DF relay system based on UAV and maximizes the instantaneous data rate by jointly optimizing beamforming and power allocation under the constraints of a single power and total power of the source node and the relay node. They proposed an effective algorithm and obtained the suboptimal solution of the problem by decomposing it into two sub-problems based on the block coordinate descent method: the beamforming optimization sub-problem with given power allocation and the power allocation sub-problem with fixed beamforming.

To sum up, the network where UAV is only used to supply for the relay with wireless energy transmission technology is not considered in the existing literature aboutthe UAV-assisted relay network. However, it is meaningful for the energy-limited relay network.

This paper attempts to optimize power allocation for energy-limited wireless relay networks, in which UAV charges the relay nodes through wireless energy transmission technology. The outage probability with/without diversity is derived. Furthermore, two optimization problems are studied for different application scenarios. The first optimization problem tries to minimize the relay probability of the system when the total power of UAV and source node is certain, to improve the reliability of the system. The second optimization problem considers that on the premise that the system meets a certain transmission rate, the total power consumption is minimized through power optimization to prolong the lifetime of energy-constrained networks. Moreover, simulation verifies the correctness of the theoretical analysis.

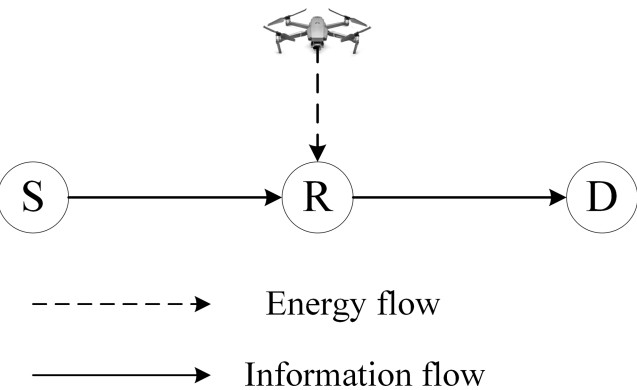

Energy flow

Information flow

**Figure 1 The illustration of UAV-assisted wireless powered relay network.** The information is transmitted from the source to the node with the energy harvested from the UAV's RF signal in the down-link.

The structure of this paper is organized as following: 'System Model' introduces the system model of UAV-assisted wireless relay network and obtains the outage probabilities; 'Power Allocation to Minimize the Outage Probability' and 'Power Distribution with a Certain Transmission Rate' formulate the optimization problem of optimal power allocation to minimize the outage probabilities under different conditions. The theoretical analysis is verified by compared with the simulation results provided in 'Simulation and Discussion'. The conclusion of the paper is presented in 'Conclusion'.

## SYSTEM MODEL

As shown in Fig. 1, the UAV-assisted energy-constrained wireless relay network is studied in this paper, which consists of a UAV node $U$, a source node $S$, an energy-constrained relay node $R$, and a destination node $D$. DF protocol is chosen by the relay node $R$ as it outperforms AF protocol. Both systems with and without diversity gain have been studied and the selection combining is picked for the information received at the destination node. The transmit power of the UAV node $U$, the source node $S$, and the energy-constrained relay node $R$ are represented by $P_U$, $P_S$, and $P_R$, respectively. The channels between nodes are statistically independent. The terrestrial channels are all Rayleigh channels, subject to frequency flat-fading. The channel gains from the source node to the relay node, between the source node and the destination node, and relay-to-destination are denoted by $h_{sr}$, $h_{sd}$, and $h_{rd}$, respectively. The channel between the UAV node and the relay node is assumed to be the line-of-sight channel.

As shown in Fig. 2, the whole DF cooperation is divided into three phases. In the first phase, the UAV transmits wireless signals to the relay node within the initial $\alpha T$ time, and the energy-constrained relay node uses wireless energy transmission technology for energy harvesting from the RF signals from UAV. Therefore, the energy collected by the relay node can be denoted by (31):

$$E_h = \frac{\beta_0}{d_U^2} P_U \eta \alpha T \tag{1}$$

**Figure 2  The schematic of the time allocation.** The schematic of time allocation during a block time T. The block time T is divided into three phase. The first phase is for energy harvested. Then the second $(1-\alpha)T/2$ is used for information transmission from source node to relay node. And the information is transmitted form relay node to destination node with the remaining time $(1-\alpha)T/2$.

where $\beta_0$ is the power attenuation factor of the line-of-sight channel; $\eta$ is the efficiency of energy collection; $\alpha$ is the time fraction of the charging process; $d_U$ isthe distance between the UAV and the relay node.

In the second phase, the next $(1-\alpha)T/2$ time, the source node broadcasts the RF signal that is needed to send, the relay node and the destination node receive the signal from the source node at the same time, and the signal received at the destination node is:

$$y_{\text{sd}}(k) = \frac{1}{\sqrt{d_{SD}}}\sqrt{P_S}h_{\text{sd}}s(k) + n_a^{[\text{sd}]}(k) + n_c^{[\text{sd}]}(k) \tag{2}$$

where $d_{\text{SD}}$ is the distance of the direct link from the source node to the destination node, $n_a^{[\text{sd}]}(k)$ is the Gaussian White Noise introduced by the receiving antenna at the destination node in the direct channel, and the variance is $\sigma_{n_a^{[\text{sd}]}}^2$, $n_c^{[\text{sd}]}(k)$ is the Gaussian white noise introduced by the conversion circuit, and the variance is $\sigma_{n_c^{[\text{sd}]}}^2$, $s(k)$ is the signal sent by the source node and it is a baseband signal after down-conversion.

Meanwhile, the signal received at the relay node is:

$$y_{sr}(k) = \frac{1}{\sqrt{d_{SR}}}\sqrt{P_S}h_{sr}s(k) + n_a^{[\text{sr}]}(k) + n_c^{[\text{sr}]}(k) \tag{3}$$

where $d_{\text{SR}}$ is the distance from the source node to the relay node, $n_a^{[\text{sr}]}(k)$ is the Gaussian white noise introduced by the receiving antenna at the relay node, and its variance is $\sigma_{n_a^{[\text{sr}]}}^2$, $n_c^{[\text{sr}]}(k)$ is the Gaussian white noise introduced by the conversion circuit at the relay node, and its variance is $\sigma_{n_c^{[\text{sr}]}}^2$.

In the third phase, that is the last $(1-\alpha)T/2$ time, the relay node decodes the received signal and then forwards the information to the destination node with the energy harvested at the first phase. So, the signal received by the destination node can be described as

$$y_{rd}(k) = \frac{1}{\sqrt{d_{RD}}}\sqrt{P_R}h_{rd}s(k) + n_a^{[rd]}(k) + n_c^{[rd]}(k) \tag{4}$$

where $n_a^{[rd]}(k)$ is the Gaussian White Noise introduced by the receiving antenna, and the variance is $\sigma_{n_a^{[rd]}}^2$. $n_c^{[rd]}(k)$ is the noise introduced by the conversion circuit, and the variance is $\sigma_{n_c^{[rd]}}^2$, $d_{RD}$ is the distance between relay node and the destination node.

According to formula (2), the instantaneous signal-to-noise ratio at the relay node can be given as

$$\gamma_{SR} = \frac{|h_{sr}|^2 P_S}{d_{SR}\sigma_{SR}^2} \tag{5}$$

where $\sigma_{SR}^2$ is the total variance of Gaussian White Noise at the relay node and $\sigma_{SR}^2 = \sigma_{n_a^{[sr]}}^2 + \sigma_{n_c^{[sr]}}^2$.

In the direct link, the instantaneous signal-to-noise ratio at the destination node is:

$$\gamma_{SD} = \frac{|h_{sd}|^2 P_S}{d_{SD}\sigma_{SD}^2} \tag{6}$$

where, $\sigma_{SD}^2$ is the total variance of Gaussian white noise at the destination node in the direct link, and $\sigma_{SD}^2 = \sigma_{n_a^{[sd]}}^2 + \sigma_{n_c^{[sd]}}^2$.

The instantaneous signal-to-noise ratio at the destination node in the two-hop link is:

$$\gamma_{RD} = \frac{2\eta P_U \beta_0 \alpha |h_{rd}|^2}{d_U^2 d_{RD}\sigma_{RD}^2(1-\alpha)} \tag{7}$$

where $\sigma_{RD}^2$ is the total variance of Gaussian white noise at the destination node in the two-hop link, $\sigma_{RD}^2 = \sigma_{n_a^{[rd]}}^2 + \sigma_{n_c^{[rd]}}^2$.

For the sake of simplicity, it is defined that, $G_0 = \frac{E[|h_{sd}|^2]}{d_{SD}\sigma_{SD}^2}$, $G_1 = \frac{E[|h_{sr}|^2]}{d_{SR}\sigma_{SR}^2}$, $G_2 = \frac{2\eta\beta_0\alpha E[|h_{rd}|^2]}{d_{RD}d_U^2(1-\alpha)\sigma_{RD}^2}$. So $\overline{\gamma_{SD}} = G_0 P_S$, $\overline{\gamma_{SR}} = G_1 P_S$, $\overline{\gamma_{RD}} = G_2 P_U$.

In the delay-tolerance system, we have $R = \log_2(1 + \gamma_0)$, where $R$ is the fixed transmission rate of the source node, $\gamma_0$ is the signal-to-noise ratio threshold for correct decoding at the destination node. Therefore, the outage probability is the probability when the signal-to-noise ratio at the destination node is less than threshold $\gamma_0$. The outage probability from $S$ to $D$ in the direct link can be calculated as

$$P_{out1} = 1 - e^{-\frac{\gamma_0}{\gamma_{SD}}} \tag{8}$$

For a two-hop DF relay network, the outage probability of the system is limited to the one-hop which is with the poor channel quality. If the direct channel is not considered, that is to say, there is no diversity gain, the outage probability at the destination node is

$$P_{out}^{DF} = 1 - e^{-\frac{\gamma_0}{\gamma_{SR}}} e^{-\frac{\gamma_0}{\gamma_{RD}}} \tag{9}$$

Then, it is obvious that the outage probability of the two-hop relay network with a diversity system can be derived as

$$P_{out2} = \left(1 - e^{-\frac{\gamma_0}{\gamma_{SR}}} e^{-\frac{\gamma_0}{\gamma_{RD}}}\right) \times \left(1 - e^{-\frac{\gamma_0}{\gamma_{SD}}}\right) \tag{10}$$

In particular, the network with diversity can achieve a smaller outage probability than the system without diversity, that is to say,

$$\left(1 - e^{-\frac{\gamma_0}{\gamma_{SR}}} e^{-\frac{\gamma_0}{\gamma_{RD}}}\right) \times \left(1 - e^{-\frac{\gamma_0}{\gamma_{SD}}}\right) \le \left(1 - e^{-\frac{\gamma_0}{\gamma_{SR}}} e^{-\frac{\gamma_0}{\gamma_{RD}}}\right) \tag{11}$$

# POWER ALLOCATION TO MINIMIZE THE OUTAGE PROBABILITY

This section studies the power allocation strategy based on the analysis of the outage probability. The power allocation will be discussed for two situations when the system is both with diversity and without diversity.

## Problem description

The goal of power allocation is to minimize the outage probability under a certain total transmit power $P_T$. Assume that the maximum transmission power of the source node is $P_{max1}$, and the maximum transmission power of the UAV is $P_{max2}$. The total transmit power $P_T$ is generally the maximum power allowed for a given data packet to be transmitted from the source node to the destination node, while $P_{max1}$ and $P_{max2}$ correspond to the maximum power that the source node and UAV can provide, respectively ($0.5 P_T < P_{max1}$, $P_{max2} \leq P_T$), then the constrained optimization problem can be expressed as:

$$
\begin{aligned}
&\min_{P_S, P_U} P_{out} \\
s.t. &\begin{cases} P_S + P_U = P_T \\ P_S \leq P_{max}, P_U \leq P_{max2} \end{cases}
\end{aligned}
\tag{12}
$$

In the above formula, the objective function is the convex function of the variables $P_S$ and $P_U$, whether there is a diversity gain or no diversity gain. The constraint function is the linear function of the optimization variables $P_S$ and $P_U$. Therefore, this is a convex optimization problem. There exists a globally optimal solution, which can be solved by Lagrange Multiplier Method.

## Without diversity

The system outage probability without diversity is shown in Eq. (9), then the optimization problem can be expressed as:

$$
\begin{aligned}
&\min_{P_S, P_U} P_{out}^{DF} = 1 - e^{-\frac{\gamma_0}{\gamma_{SR}}} e^{-\frac{\gamma_0}{\gamma_{RD}}} \\
s.t. &\begin{cases} P_S + P_U = P_T \\ P_S \leq P_{max}, P_U \leq P_{max2} \end{cases}
\end{aligned}
\tag{13}
$$

In the formula above, minimizing the objective function is equivalent to maximizing the expression $-\frac{\gamma_0}{\gamma_{SR}} - \frac{\gamma_0}{\gamma_{RD}}$, then the optimization problem is simplified to be

$$
\begin{aligned}
&\max_{P_S, P_U} \left( -\frac{\gamma_0}{\gamma_{SR}} - \frac{\gamma_0}{\gamma_{RD}} \right) \\
s.t. &\begin{cases} P_S + P_U = P_T \\ P_S \leq P_{max}, P_U \leq P_{max2} \end{cases}
\end{aligned}
\tag{14}
$$

To solve the problem, the Lagrange Multiplier Method is picked. If the second restriction is neglected, the objective function is constructed as:

$$
F_1(P_S, P_U, \eta) = -\frac{\gamma_0}{\gamma_{SR}} - \frac{\gamma_0}{\gamma_{RD}} + \eta(P_S + P_U - P_T)
\tag{15}
$$

According to the necessary for the existence of extreme values, it can be obtained

$$\begin{cases} P_S^* = \dfrac{\sqrt{G_2}P_T}{\sqrt{G_1}+\sqrt{G_2}} \\ P_U^* = \dfrac{\sqrt{G_1}P_T}{\sqrt{G_1}+\sqrt{G_2}} \end{cases} \tag{16}$$

As the value of the power given in Eq. (16) may be out of range, the second constraint in Eq. (14) is combined to determine the optimal power. Moreover, if $P_S^*$ is out of bounds, the optimal power allocation can be obtained as

$$\begin{cases} P_S^* = P_{\max} \\ P_U^* = P_T - P_{\max 1} \end{cases} \tag{17}$$

Conversely, if $P_U^*$ crosses the boundary, the optimal power can be derived by

$$\begin{cases} P_U^* = P_{\max} \\ P_S^* = P_T - P_{\max} \end{cases} \tag{18}$$

## With diversity

Since the system outage probability with diversity gain is given in Eq. (10), the optimization problem, in this case, can be built as:

$$\min_{P_S,P_U} P_{out2} = \left(1 - e^{-\frac{\gamma_0}{\gamma_{SR}}} e^{-\frac{\gamma_0}{\gamma_{RD}}}\right) \times \left(1 - e^{-\frac{\gamma_0}{\gamma_{SD}}}\right)$$
$$s.t. \begin{cases} P_S + P_U = P_T \\ P_S \le P_{\max} \prime P_U \le P_{\max 2} \end{cases} \tag{19}$$

Similarly, using Lagrange Method and ignoring the second constraint in Eq. (19), then the simplified optimization function is:

$$F_2(P_S,P_U,\mu) = 1 - e^{-\frac{\gamma_0}{\gamma_{SD}}} - e^{-\frac{\gamma_0}{\gamma_{SR}}} e^{-\frac{\gamma_0}{\gamma_{RD}}} + e^{-\frac{\gamma_0}{\gamma_{SR}}} e^{-\frac{\gamma_0}{\gamma_{RD}}} e^{-\frac{\gamma_0}{\gamma_{SD}}} + \mu(P_S + P_U - P_T) \tag{20}$$

The equations that the extreme values should be satisfied are as follows,

$$\begin{cases} -\dfrac{\gamma_0}{G_0 P_S^2} e^{-\frac{\gamma_0}{G_0 P_S}} - \dfrac{\gamma_0}{G_1 P_S^2} e^{-\frac{\gamma_0}{G_1 P_S}} e^{-\frac{\gamma_0}{G_2 P_U}} + e^{-\frac{\gamma_0}{G_0 P_S}} e^{-\frac{\gamma_0}{G_1 P_S}} e^{-\frac{\gamma_0}{G_2 P_U}} \left(\dfrac{\gamma_0}{G_0 P_S^2} + \dfrac{\gamma_0}{G_1 P_S^2}\right) + \mu = 0 \\ -\dfrac{\gamma_0}{G_2 P_U^2} e^{-\frac{\gamma_0}{G_2 P_U}} e^{-\frac{\gamma_0}{G_1 P_S}} + \dfrac{\gamma_0}{G_2 P_U^2} e^{-\frac{\gamma_0}{G_2 P_U}} e^{-\frac{\gamma_0}{G_1 P_S}} e^{-\frac{\gamma_0}{G_0 P_S}} + \mu = 0 \\ P_S + P_U - P_T = 0 \end{cases} \tag{21}$$

The transmission power of the source node and the UAV can be obtained by solving the formula above and the expression that the optimal values need to satisfy is

$$\begin{cases} P_S^* = \sqrt{\gamma_0\left(\dfrac{1}{G_0} + \dfrac{1}{G_1}\right)\left(\dfrac{\gamma_0}{G_0 P_S^2} e^{\frac{\gamma_0}{G_1 P_S}} e^{\frac{\gamma_0}{G_2 P_U}} + \dfrac{\gamma_0}{G_1 P_S^2} e^{\frac{\gamma_0}{G_0 P_S}} - \dfrac{\gamma_0}{G_2 P_U^2} e^{\frac{\gamma_0}{G_0 P_S}} - \dfrac{\gamma_0}{G_2 P_U^2}\right)^{-1}} \\ P_U^* = P_T - P_S^* \end{cases} \tag{22}$$

where $P_S^*$ and $P_U^*$ is the optimal allocation power.

Although the equation in (22) is not a closed-form solution about $P_S^*$ and $P_U^*$, its optimal solution can be obtained by successive approximation. It should be noted that, if $P_S^* > P_{max1}$ during the iteration process appears, the iteration is aborted to ensure the KKT conditions. Thus, the optimal value of $P_S^*$ is $P_{max1}$, and the optimal power allocation is:

$$
\begin{cases}
P_S^* = P_{max} \\
P_U^* = P_T - P_{max}
\end{cases}
\tag{23}
$$

Similarly, if the power value of the UAV exceeds $P_{max2}$ in the iterative process, it is reduced to $P_{max2}$, and the reduced power is allocated to the source node. Therefore, the optimal power distribution is given by:

$$
\begin{cases}
P_U^* = P_{max2} \\
P_S^* = P_T - P_{max2}
\end{cases}
\tag{24}
$$

## POWER DISTRIBUTION WITH A CERTAIN TRANSMISSION RATE

The section studies the power allocation strategy when the outage probability of the system meets a certain threshold based on the analysis of system outage probability. The power will be allocated for the cases with diversity and without diversity, respectively.

### Problem description

The target of power allocation is to minimize the total power through power allocation under the condition that the outage probability at the destination node guarantees a certain threshold condition.

It is supposed that the threshold of outage probability is $\rho_0$, then the constrained optimization problem can be expressed as:

$$
\min_{P_S, P_U} P_S + P_U
$$
$$
s.t. P_{out} \leq \rho_0
\tag{25}
$$

Because the constraints are non-convex, the problem above is not a convex optimization. It can be solved by the Lagrange multiplier method.

### Without diversity

Substituting Eq. (9) into the expression above, the constrained minimization problem for a system without diversity can be obtained as

$$
\min_{P_S, P_U} P_S + P_U
$$
$$
s.t. 1 - e^{-\frac{\gamma_0}{\gamma_{SR}}} e^{-\frac{\gamma_0}{\gamma_{RD}}} - \rho_0 \leq 0
\tag{26}
$$

The Lagrange multiplier method is used for optimization, and the constructor is:

$$
L(P_S, P_U, \theta) = P_S + P_U + \theta \left( 1 - e^{-\frac{\gamma_0}{\gamma_{SR}}} e^{-\frac{\gamma_0}{\gamma_{RD}}} - \rho_0 \right)
\tag{27}
$$

The optimal value satisfies the equation as follows:

$$
\begin{cases}
\dfrac{\partial L(P_S, P_U, \theta)}{\partial P_S} = 1 + \mu\left(-\dfrac{\gamma_0}{G_1 P_S^2} e^{-\frac{\gamma_0}{G_2 P_U}} e^{-\frac{\gamma_0}{G_1 P_S}}\right) = 0 \\
\dfrac{\partial L(P_S, P_U, \theta)}{\partial P_U} = 1 + \mu\left(-\dfrac{\gamma_0}{G_2 P_U^2} e^{-\frac{\gamma_0}{G_2 P_U}} e^{-\frac{\gamma_0}{G_1 P_S}}\right) = 0 \\
\dfrac{\partial L(P_S, P_U, \theta)}{\partial \mu} = 1 - e^{-\frac{\gamma_0}{G_2 P_U}} e^{-\frac{\gamma_0}{G_1 P_S}} - \rho_0 = 0
\end{cases}
\tag{28}
$$

From the first two formulas of the equations above, the relationship between $P_S$ and $P_U$ can be obtained as $P_S = (G_2/G_1)^{1/2} P_U$, and the optimal power can be expressed as

$$
\begin{cases}
P_S = -\dfrac{\gamma_0}{\ln(1-\rho_0)}\left(\dfrac{1}{\sqrt{G_1 G_2}} + \dfrac{1}{G_1}\right) \\
P_U = -\dfrac{\gamma_0}{\ln(1-\rho tensor*[0])}\left(\dfrac{1}{\sqrt{G_1 G_2}} + \dfrac{1}{G_2}\right)
\end{cases}
\tag{29}
$$

## With diversity

By substituting Eq. (10) into (25), the constrained minimization problem for a system with diversity can be expressed as

$$
\min_{P_S, P_U} P_S + P_U
$$
$$
s.t. \left(1 - e^{-\frac{\gamma_0}{\gamma_{SR}}} e^{-\frac{\gamma_0}{\gamma_{RD}}}\right) \times \left(1 - e^{-\frac{\gamma_0}{\gamma_{SD}}}\right) - \rho_0 \leq 0
\tag{30}
$$

Use Lagrange Multiplier Method to optimize, the constructor is:

$$
L(P_S, P_U, \omega) = P_S + P_U + \omega\left[\left(1 - e^{-\frac{\gamma_0}{\gamma_{SR}}} e^{-\frac{\gamma_0}{\gamma_{RD}}}\right) \times \left(1 - e^{-\frac{\gamma_0}{\gamma_{SD}}}\right) - \rho_0\right]
\tag{31}
$$

According to the necessary existence conditions of extreme values:

$$
\begin{cases}
\dfrac{\partial L(P_S, P_U, \omega)}{\partial P_S} = 1 + \mu\dfrac{\partial}{\partial P_S}\left[\left(1 - e^{-\frac{\gamma_0}{\gamma_{SR}}} e^{-\frac{\gamma_0}{\gamma_{RD}}}\right) \times \left(1 - e^{-\frac{\gamma_0}{\gamma_{SD}}}\right)\right] = 0 \\
\dfrac{\partial L(P_S, P_U, \omega)}{\partial P_U} = 1 + \mu\dfrac{\partial}{\partial P_U}\left[\left(1 - e^{-\frac{\gamma_0}{\gamma_{SR}}} e^{-\frac{\gamma_0}{\gamma_{RD}}}\right) \times \left(1 - e^{-\frac{\gamma_0}{\gamma_{SD}}}\right)\right] = 0 \\
\left(1 - e^{-\frac{\gamma_0}{\gamma_{SR}}} e^{-\frac{\gamma_0}{\gamma_{RD}}}\right) \times \left(1 - e^{-\frac{\gamma_0}{\gamma_{SD}}}\right) - \rho_0 = 0
\end{cases}
\tag{32}
$$

From the first two constraints above, the relationship between $P_S$ and $P_U$ can be obtained as

$$
\dfrac{1}{G_0 P_S^2} e^{-\frac{\gamma_0}{G_0 P_S}}\left(e^{-\frac{\gamma_0}{G_1 P_S}} e^{-\frac{\gamma_0}{G_2 P_U}} - 1\right) = e^{-\frac{\gamma_0}{G_1 P_S}} e^{-\frac{\gamma_0}{G_2 P_U}}\left(\dfrac{1}{G_2 P_U^2} - \dfrac{1}{G_1 P_S^2}\right)\left(e^{-\frac{\gamma_0}{G_0 P_S}} - 1\right)
\tag{33}
$$

By simultaneous the third formula in (32) and the formula in (33), we can obtain the optimal power.

Considering the computation complexity is high, we tried to use the relationship between $P_S$ and $P_U$ to give a relatively simple solution. First of all, as the constraint in Eq. (33) is valid, we could take the equal sign. When the feasible region of $P_S$ needs to meet certain conditions, $P_U$ can be expressed as a function of $P_S$ as

$$
P_U = \dfrac{-\gamma_0}{G_2\left[\dfrac{\gamma_0}{G_1 P_S} + \ln\left(1 - \dfrac{\rho_0}{1 - e^{-\frac{\gamma_0}{G_0 P_S}}}\right)\right]}
\tag{34}
$$

On the one hand, as the entire cooperation process only works when the signal-to-noise ratio of the direct channel is greater than the threshold $\rho_0$, there is

$$1 - e^{-\frac{\gamma_0}{\gamma_{SD}}} > \rho_0 \tag{35}$$

Solving the formula above, we can get $P_S < P_{th1}$, and $P_{th1} = -\gamma_0/\ln(1-\rho_0)G_0$. On the other hand, the transmission power of the UAV is non-negative, then it can be derived according to the (32), as

$$\left(1 - e^{-\gamma_0/G_1}\right)\left(1 - e^{-\gamma_0/G_0}\right) < \rho_0 \tag{36}$$

The values which satisfy $P_S < P_{th2}$ can be obtained by numerical methods according to the Eq. (34). Thus, the two-dimensional problem in the Eq. (30) can be simplified to a one-dimensional optimization problem according to Eqs. (34), (35) and (36). The problem can be remodeled as

$$\min_{P_S, P_U} \widetilde{f}(P_S) = P_S + \frac{-\gamma_0}{G_2\left[\frac{\gamma_0}{G_1 P_S} + \ln\left(1 - \frac{\rho_0}{1 - e^{-\frac{\gamma_0}{G_0 P_S}}}\right)\right]} \tag{37}$$

$$s.t. \, P_{th1} < P_S < P_{th2}$$

To obtain the minimum total power when the system satisfies the transmission conditions, the derivative of $\widetilde{f}(P_S)$ needs to be obtained, and the equation that optimal $P_S^*$ needs to satisfy can be expressed as

$$\left[\gamma_0 q_1(P_S^*)\right] / \left[G_2 q_2(P_S^*)\right] = -1 \tag{38}$$

where, $q_1(P_S^*) = \dfrac{-\gamma_0}{(P_S^*)^2}\left[\dfrac{\rho_0 e^{-\frac{\gamma_0}{G_0}}}{G_0\left(1 - e^{-\frac{\gamma_0}{G_0}}\right)\left(1 - e^{-\frac{\gamma_0}{G_0}} - \rho_0\right)} + \dfrac{1}{G_1}\right]$ and

$q_2(P_S^*) = \left[\dfrac{\gamma_0}{G_1} + \ln\left(1 - \dfrac{\rho_0}{1 - e^{-\frac{\gamma_0}{G_0}}}\right)\right]^2.$

It should be noted that $P_U$ is a monotonically decreasing function of $P_S$ within the feasible range. Therefore, the optimal transmit power $P_S^*$ of the source node can be easily obtained by numerical methods if an initial value of $P_S$ is given. Then the optimal transmission power of the UAV can be obtained by substituting $P_S^*$ into the Eq. (34).

## Simulation

The theoretical derivation in the previous is simulated in this section, and the correctness of the theory is verified. Moreover, the influence of the signal-to-noise ratio threshold on the optimal power allocation strategy is further discussed under different diversity reception modes. The simulation parameters are set as follows: the corresponding average power gains are given as $G_0 = 1$, $G_1 = 5$, $G_2 = 0.5$, respectively and threshold of the signal-to-noise ratio is set as $\gamma_0 = 5$dB.

Figure 3 shows the optimal power allocation strategy for a network with diversity when the signal-to-noise ratio at the destination node is no less than the preset threshold.

We simulate the third equation in Eqs. (32) and (33) (the curve and virtual real line in Fig. 3 respectively) and find the intersection of the two lines which is the value of optimal

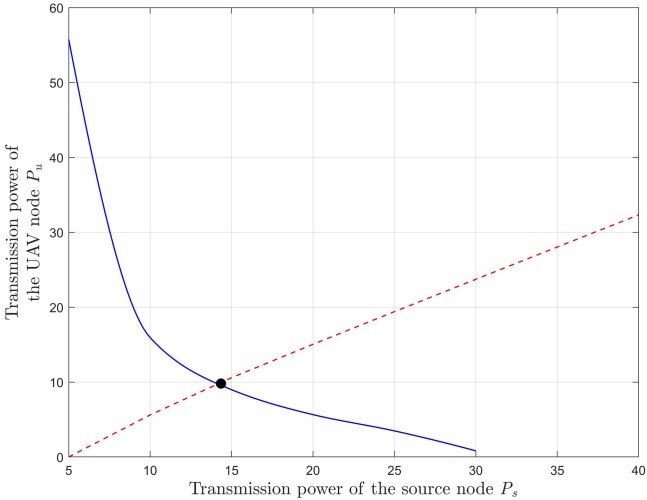

**Figure 3 Calculation of the optimal power under the outage probability threshold.** We simulate the third equation in Eqs. (30) and (31) (the curve and virtual real line respectively) and find the intersection of the two lines which is the value of optimal power. To reduce the computational complexity, the optimal power can also be obtained by simultaneous Eqs. (34) and (37), as shown by the black dot in the figure.

power. To reduce the computational complexity, the optimal power can also be obtained by simultaneous Eqs. (34) and (37), as shown by the black dot in the figure. We find that the same optimal power can be obtained by the two methods, which verifies the correctness of the theoretical derivation.

The optimal transmission power for both with/without diversity is depicted in Figs. 4 and 5, which vary with the outage probability threshold, respectively. The simulation value range of the outage probability threshold is from $10^{-3}$ to 0.1. It can be seen that the optimal transmission power required by the network is significantly reduced since the outage probability constraint is relaxed. In addition, the optimal power with diversity is significantly less than it without diversity for the same outage probability threshold. This is because there is an extra direct link from the source node to the destination node in the diversity system, which strengthens the SNR at the destination node.

## SIMULATION AND DISCUSSION

This section analyzes the performance of the system and verifies the correctness of the theoretical analysis by simulation. Unless otherwise specified, the parameters of the system during the simulation are set as follows: the distance from the source node to the destination node is 1.5, the distance from the source node to the relay node and from the relay node to the destination node is 1, and the distance from the UAV node to the relay node is 0.2. The mean values of the channel parameters $h2$ sr, $h2$ sd and $h2$ rd are all units. For simplicity, it is assumed that the noise variables at the relay node and the destination node are similar, $\sigma_{SR}^2 = \sigma_{RD}^2 = \sigma_{SD}^2 = 0.3$. The charging time ratio is 0.3, the line-of-sight channel attenuation factor is 2, and the line-of-sight parameter is $-30$ dB. The charging efficiency is 1, and the threshold of the signal-to-noise ratio is 3.

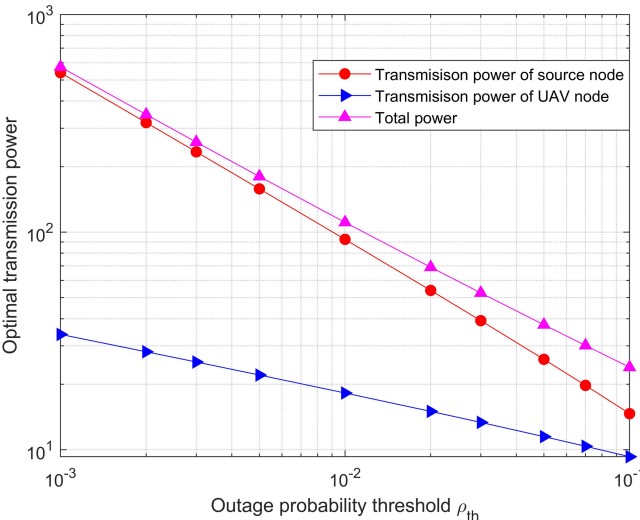

**Figure 4  Optimal power versus the outage probability threshold for the system with diversity.** The optimal transmission power required by the network is significantly reduced since the outage probability constraint is relaxed.

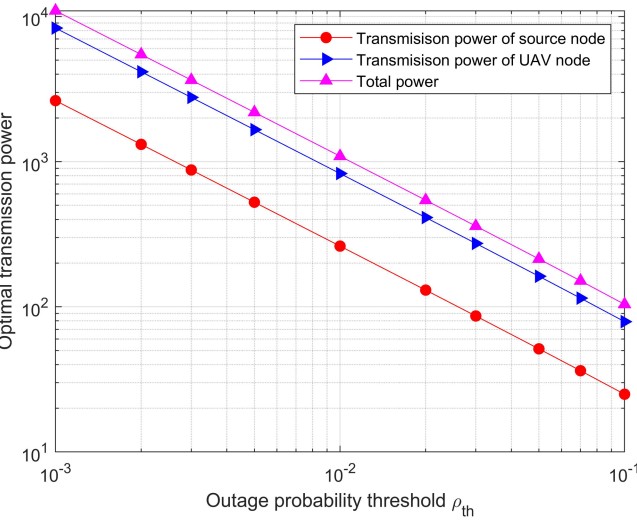

**Figure 5  Optimal power versus the outage probability threshold for the system without diversity.** It can be seen that the optimal transmission power required by the network is significantly reduced since the outage probability constraint is relaxed. In addition, the optimal power with diversity is significantly less than it without diversity for the same outage probability threshold.

Figure 6 shows the outage probability performance curves of the optimal power allocation algorithm proposed in this paper and the average power allocation algorithm when different relay cooperation modes (with diversity and without diversity) are selected in the UAV-assisted wireless network. It can be seen that the outage probability decreases with the increase of total power, and the performance of the system outage probability is always

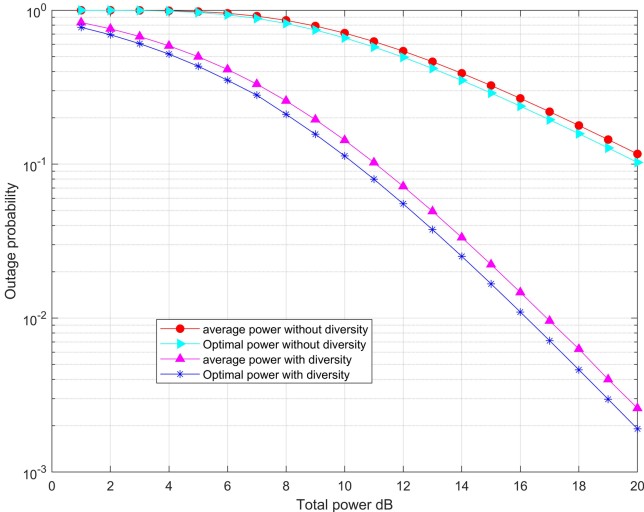

**Figure 6  Minimization of the outage probability versus the total power threshold for the system with-/without diversity.** The outage probability decreases with the increase of total power, and the performance of the system outage probability is always more superior lower when the cooperative mode with diversity gain is adopted, which means that it can obtain the same outage requirements with less energy.

more superior lower when the cooperative mode with diversity gain is adopted, which means that it can obtain the same outage requirements with less energy. This is because the increased direct channel from the source node to the destination node increases the signal-to-noise ratio at the destination node, thus reducing the outage probability of the system. The simulation results are consistent with the previous theoretical analysis.

Figure 7 compares the outage probability of the system when the diversity and without diversity reception modes are adopted destination nodes respectively under different time ratios. Obviously, the outage probability decreases as the energy collection time ratio continuously increases which enables the available energy in the system to increase. It can be seen from the figure that the optimal power allocation algorithm proposed in this paper can significantly reduce the outage probability of the system compared with the average power allocation algorithm. In particular, the outage probability of the system without diversity gain is 1 in the case that the time ratio of energy collection is 0. The reason is that the energy available at the limited relay node to forward the information from the source node to the destination, the signal-to-noise ratio at the destination node is all less than the threshold. However, the outage probability of the system is not 1 where the system is with diversity due to the existence of a direct link.

To investigate the throughput of the system, the throughput of the time split is described in Fig. 8 both with/without diversity, respectively. It can be seen from Fig. 8 that the optimal power allocation algorithm proposed in this paper is better than the average power allocation algorithm. Moreover, the system with diversity can obtain greater throughput than the system without diversity. For the system without diversity, the throughput increases from 0 to a maximum value and then decreases to 0 with the increase of $\alpha$. This is because the relay node collects very little energy result in a very high outage probability

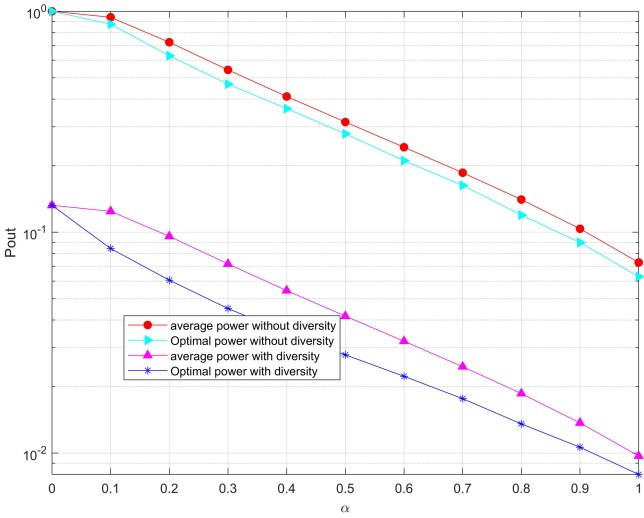

**Figure 7** **Minimization of the outage probability versus under the total power threshold.** The outage probability of the system is compared when the diversity and without diversity reception modes are adopted destination nodes respectively under different time ratios. Obviously, the outage probability decreases as the energy collection time ratio continuously increases which enables the available energy in the system to increase.

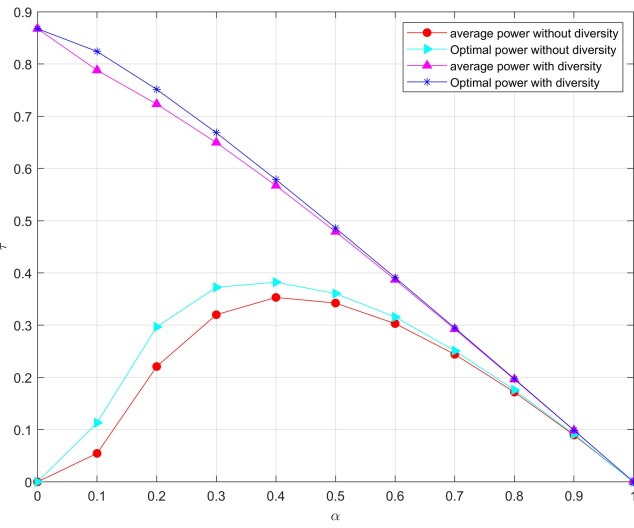

**Figure 8** **Maximization of the outage probability versus $\alpha$ under the total power threshold.** The optimal power allocation algorithm proposed in this paper is better than the average power allocation algorithm.

of the system and then a small throughput when $\alpha$ is very small. Nevertheless, with the continuous increase of $\alpha$, the energy input to the system increases, and the throughput of the system increases.

Then, the outage probability of the system decreases, and the time for information transmission decreases, so the throughput of the system decreases as the $\alpha$ continues to

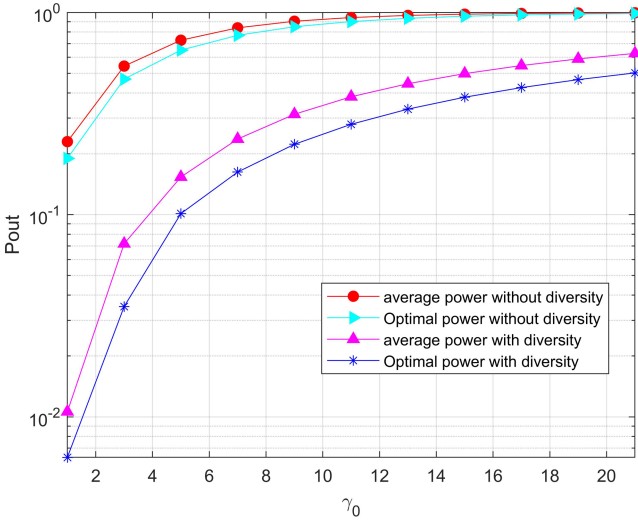

**Figure 9** **Minimization of the outage probability versus $\gamma_0$ under the total power threshold.** The outage probability continues to increase together with the increase of the SNR threshold, as the SNR threshold at the destination node is more and more difficult to reach.

increase. Particularly, the throughput of the system is decreasing, and the outage probability is decreasing meanwhile for the system with diversity as the energy collection time of relay nodes is continue to increase. Therefore, an appropriate $\alpha$ should be chosen to make a compromise between the outage probability and the throughput.

The outage probability of the system changes with the signal-to-noise ratio threshold at the destination node with/without diversity is shown in Fig. 9, respectively. The outage probability continues to increase together with the increase of the SNR threshold, as the SNR threshold at the destination node is more and more difficult to reach. It can be seen from the figure that the optimal power allocation algorithm proposed in this paper can reduce the system outage probability compared with the average power allocation method, especially for the system with diversity. Moreover, the outage probability remains to be 1 when the signal-to-noise ratio threshold is greater than 15 in the system without diversity, which is mainly because the signal-to-noise ratio threshold at the destination node is too large to achieve. Similarly, there also exists a certain range that the outage probability of the system will all be 1 as the SNR threshold at the destination node continues to increase in the system with diversity.

The system outage probability varies with the distance between the UAV and the relay node is revealed in reveal Fig. 10 in the network with/without diversity, respectively. It can be observed from the figure that the outage probability increases as the distance of the UAV to the relay node is continuing to increase. The main reason is that the energy collected by the relay node from the UAV's RF signal becomes less and the available energy of the system decreases. Obviously, the outage probability in the system with diversity is significantly lower than that in the without diversity reception. In addition, the optimal power allocation algorithm proposed in this paper can reduce the outage probability of

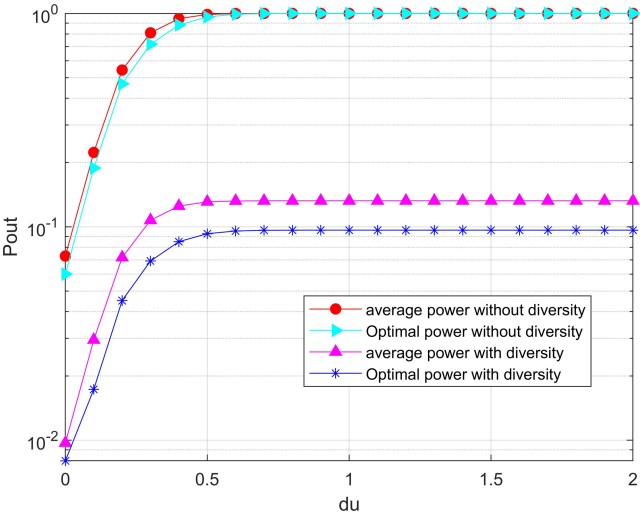

**Figure 10 Minimization of the outage probability versus du under the total power threshold.** It can be observed from the figure that the outage probability increases as the distance of UAV to relay node is continuing to increase. The main reason is that the energy collected by the relay node from the UAV's RF signal becomes less and the available energy of the system decreases.

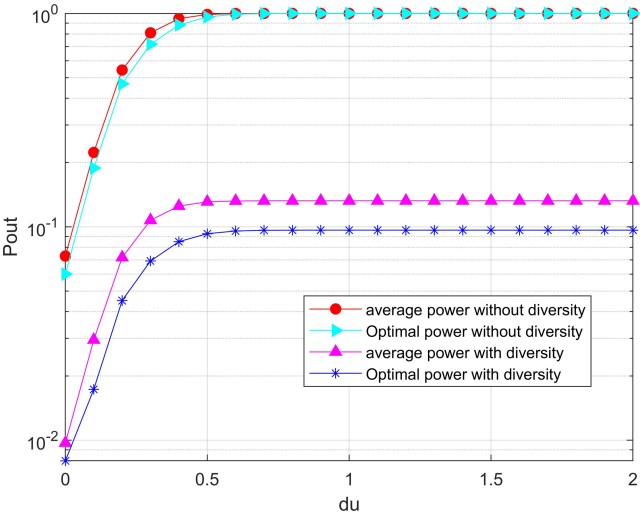

the system compared with the average power allocation algorithm. Moreover, the outage probability remains to be 1 when the distance from the UAV to the relay is greater than 0.6 in the system without diversity. This is because the energy collected by the relay node is too small to support the signal-to-noise at the destination node to reach the threshold with the increased charging distance.

Figure 11 shows the change of the system outage probability with the line-of-sight channel attenuation factor $\beta_0$ in the system with/without diversity, respectively. It can be seen that the relay node could harvest more energy from the RF signal of the UAV with the continuous increase of $\beta_0$. Thus, the outage probability decreases as the available energy of the system increases. Obviously, the outage probability in the system with diversity is significantly lower than it is without diversity. Similarly, the optimal power allocation algorithm proposed in this paper can reduce the outage probability of the system compared with the average power allocation algorithm. Moreover, the system outage probability decreases slowly with the continuous increase of $\beta_0$ when $\beta_0$ is greater than a certain value since other parameters in the system are instead of the harvested energy as the main factor to affect the outage probability.

## CONCLUSIONS

This paper studies the optimal power allocation strategy in wireless networks where the energy-constrained relay nodes through RF energy from the UAVs. Firstly, the system outage probability with and without diversity gain is derived. Then, an optimal power allocation strategy for decoding and forwarding cooperative systems with diversity under outage probability constraints is proposed in detail. On the one hand, since the optimization problem with constraints is nonconvex for the network with diversity, the conventional

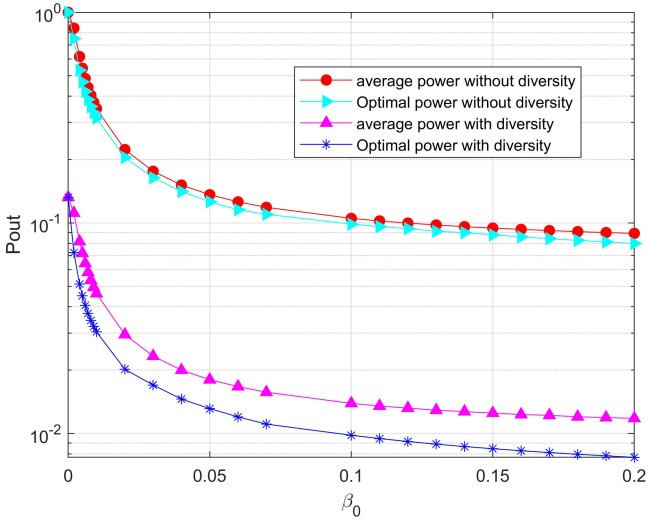

**Figure 11** **Minimization of the outage probability versus $\beta_0$ under the total power threshold.** The relay node could harvest more energy from the RF signal of the UAV with the continuous increase of $\beta_0$. Thus, the outage probability decreases as the available energy of the system increases.

convex optimization method could not be adopted to handle it. Therefore, an effective method is raised to reduce the computational complexity by using the relationship between the power of the source node and the relay node. A closed-form expression of the optimal power allocation strategy is acquired for the network without diversity. On the other hand, this paper attempts to improve the network quality of service by minimizing the outage probability under the total power constraint. As the optimization problems established are convex for both situations that whether the network is with diversity or without diversity, the Lagrange multiplier method is used to solve them. Newton iterative method is adopted to solve the system with diversity, and the analytical formula of optimal power allocation can be directly derived for the systems without diversity. Finally, simulation experiments verify the effectiveness of the optimization method and the correctness of the theoretical analysis.

### Nomenclature

| | |
|---|---|
| $\alpha$ | time ratio of energy collection |
| $\eta$ | the efficiency of energy collection |
| $\beta_0$ | the power attenuation factor of the line-of-sight channel |
| $P_U$ | the transmit power of UAV |
| $d_U$ | the distance between the UAV and the relay node |
| $d_{SD}$ | the distance from the source node to the destination node |
| $s(k)$ | the signal sent by the source node |
| $h_{sd}$ | the channel gain between the source and the destination |
| $P_S$ | the transmit power of the source node |
| $n_a^{[sd]}(k)$ | the noise introduced by the receiving antenna at destination in the direct link |
| $n_c^{[sd]}(k)$ | the noise introduced by the conversion circuit at destination in the direct link |

| | |
|---|---|
| $n_a^{[sr]}(k)$ | the noise introduced by the receiving antenna at the relay node |
| $n_c^{[sr]}(k)$ | the noise introduced by the conversion circuit at the relay node |
| $n_a^{[rd]}(k)$ | the noise introduced by the receiving antenna at the destination in the link from the relay node to the destination node |
| $n_c^{[rd]}(k)$ | the noise introduced by the conversion circuit at destination in the link from the relay node to destination node |
| $h_{\text{sd}}$ | the channel gains from the source node to the destination node |
| $h_{sr}$ | the channel gains from the source node to the destination node |
| $h_{rd}$ | the channel gains from the source node to the destination node |
| $y_{\text{sd}}(k)$ | the signal received at the destination node in the direct link |
| $y_{sr}(k)$ | the signal received at the relay node |
| $y_{rd}(k)$ | the signal received at the destination node in the backward link |
| $\gamma_{SR}$ | the signal-to-noise ratio at the relay node |
| $\gamma_{SD}$ | the signal-to-noise ratio at the relay node in the direct link |
| $\gamma_{RD}$ | the signal-to-noise ratio at the relay node in the two-hop link |
| $\sigma_{SD}^2$ | the total variance of Gaussian white noise at the destination node in the direct link |
| $\sigma_{RD}^2$ | the total variance of Gaussian white noise at the destination node in the two-hop link |
| $G_0$ | $\frac{\mathrm{E}\left[|h_{sd}|^2\right]}{d_{SD}\sigma_{SD}^2}$ |
| $G_1$ | $\frac{\mathrm{E}\left[|h_{sr}|^2\right]}{d_{SR}\sigma_{SR}^2}$ |
| $G_2$ | $\frac{2\eta\beta_0\alpha\mathrm{E}\left[|h_{rd}|^2\right]}{d_{RD}d_U^2(1-\alpha)\sigma_{RD}^2}$ |
| $\gamma_0$ | the signal-to-noise ratio threshold |
| $P_{out1}$ | the outage probability from $S$ to $D$ in the direct link |
| $P_{out2}$ | the outage probability with diversity |
| $P_{out}^{\mathrm{DF}}$ | the outage probability without diversity |
| $P_{\mathrm{T}}$ | the total transmit power |
| $P\text{max}1$ | ] the maximum transmission power of the source node |
| $P\text{max}1$ | the maximum transmission power of the UAV |
| $P_S^*$ | the optimal transmission power of the source node |
| $P_U^*$ | the optimal transmission power of the UAV |

### Funding

This research was supported by the National Natural Science Foundation of China under Grant 61842103, 61871351, and 61801437 and by the Science and Technology Foundation of State Key Laboratory of Electronic Testing Technology under Grant 6142001180410 and by the Technological Innovation Foundation of the Higher Education Institutions of Shanxi Province, China under Grant 2020L0301 and 2020L0389 and by the Poverty-relief Foundation of Shanxi Province under Grant 2020FP-11 and by Scientific and Technological Innovation Foundation of Higher Education Institutions in Shanxi Province, China under Grant 18005520 and by Science Foundation of North University of China under Grant

XJJ201927. There was no additional external funding received for this study. The funders had no role in study design, data collection and analysis, decision to publish, or preparation of the manuscript.

### Grant Disclosures

The following grant information was disclosed by the authors:

National Natural Science Foundation of China: 61842103, 61871351, 61801437.

Science and Technology Foundation of State Key Laboratory of Electronic Testing Technology: 6142001180410.

Technological Innovation Foundation of the Higher Education Institutions of Shanxi Province, China: 2020L0301, 2020L0389.

Poverty-relief Foundation of Shanxi Province: 2020FP-11.

Scientific and Technological Innovation Foundation of Higher Education Institutions in Shanxi Province, China: 18005520.

Science Foundation of North University of China: XJJ201927.

### Competing Interests

The authors declare there are no competing interests.

### Author Contributions

- Jing Yan conceived and designed the experiments, performed the experiments, analyzed the data, performed the computation work, prepared figures and/or tables, authored or reviewed drafts of the paper, and approved the final draft.
- Xuefeng Deng analyzed the data, authored or reviewed drafts of the paper, and approved the final draft.
- Jihua Liu analyzed the data, prepared figures and/or tables, and approved the final draft.
- Liming Wang conceived and designed the experiments, authored or reviewed drafts of the paper, and approved the final draft.

### Data Availability

The source code is available in the Supplementary File.

### Supplemental Information

Supplemental information for this article can be found online at http://dx.doi.org/10.7717/peerj-cs.864#supplemental-information.

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
