# Peer review of "Optimal power allocation for a wireless cooperative network with UAV"

_PeerJ Computer Science, doi:10.7717/peerj-cs.864_

## Round 0.1 · original submission · Major Revisions

The paper is not ready for publication. A revision is needed to fix some issues. Please provide a detailed point-to-point response letter along with the revision. Thanks.

Reviewer 1 ·

Basic reporting

The paper studies the optimal power allocation strategy in wireless networks where the energy-constrained relay nodes through RF energy from the UAVs. The addressed problem is interesting and the proposed solution seems effective. However, there are some weaknesses in the paper that the authors must particularly pay attention and handle:
(1) (Line 461 on page 15) The format of references is inconsistent, for example, the caption case of a reference article is inconsistent.
(2) The logic and English usage of this paper are poor. The quality of written English should be improved.

Experimental design

The paper refers to the need to find a suitable α to make a compromise between the outage probability and the throughout, is it necessary to give the appropriate α in the form of simulation?

Validity of the findings

no comment

Additional comments

(1) (Line 2 on page 1) "UAV cooperative wireless DF relay network." What is DF?
(2) The expression of the formula in this paper is wrong, and the physical meaning of some letters is unclear.
(3) (Line 224 on page 7 and line 242 on page 8) Why the second level headings B and C are the same?

Annotated reviews are not available for download in order to protect the identity of reviewers who chose to remain anonymous.

Reviewer 2 ·

Basic reporting

Yes

Experimental design

Yes

Validity of the findings

Yes

Additional comments

1. In the system model, why choose DF protocol instead of AF protocol? Any justification.

2. Why Rayleigh channel, especially when Los link exists.

3. UAV itself has very limited energy, so it will very impractical to use it as an energy source to power the relay node. The authors must provide substantial evidence to show the effectiveness.

4. What are the computation complexity of solutions used to solve the identified optimization problems?

5. Please provide more baseline strategies used for baselines in simulation.

6. Presentation:
(1) Undefined abbreviation in abstract "DF"
(2) Too many typos and grammar problems.
(3) Please put the figures at appropriate places instead of at the end of the manuscript.
(4) Provide a table to include all parameters to better help the reviewer understand.

---

## Round 0.2 · Minor Revisions

A minor revision is needed before further consideration. I look forward to receiving your revised version.

Reviewer 2 ·

Basic reporting

Good

Experimental design

Good

Validity of the findings

Good

Additional comments

Thanks for addressing my comments. Here, more justifications are needed for the following two questions:

(1) Why Rayleigh channel, especially when Los link exists. Since LOS links, the Ricean channel will be more suitable instead of Rayleigh. After reading the authors' feedback, I am not still convinced about the use of the Rayleigh channel.
(2) Instead of only text, could the authors provide some simulation results for the baseline strategy.
(3) Please insert the table of all parameters somewhere in the paper instead of listing it only in the response letter.

---

## Round 0.3 · accepted · Accept

The reviewers' comments have been addressed. I recommend it for publication.